# A Pseudo-Label Method for Coarse-to-Fine Multi-Label Learning with Limited Supervision

**Cheng-Yu Hsieh**[*][†]
r05922048@ntu.edu.tw

**Miao Xu**[*]
miao.xu@riken.jp

**Gang Niu**[*]
gang.niu@riken.jp

**Hsuan-Tien Lin**[†]
htlin@csie.ntu.edu.tw
[†]National Taiwan University

**Masashi Sugiyama**[*◇]
sugi@k.u-tokyo.ac.jp
[*]RIKEN Center for Advanced Intelligence Project
[◇]The University of Tokyo

## Abstract

The goal of multi-label learning (MLL) is to associate a given instance with its relevant labels from a set of concepts. Previous works of MLL mainly focused on the setting where the concept set is assumed to be fixed, while many real-world applications require introducing new concepts into the set to meet new demands. One common need is to refine the original coarse concepts and split them into finer-grained ones, where the refinement process typically begins with limited labeled data for the finer-grained concepts. To address the need, we propose a special weakly supervised MLL problem that not only focuses on the situation of limited fine-grained supervision but also leverages the hierarchical relationship between the coarse concepts and the fine-grained ones. The problem can be reduced to a multi-label version of negative-unlabeled learning problem using the hierarchical relationship. We tackle the reduced problem with a meta-learning approach that learns to assign pseudo-labels to the unlabeled entries. Experimental results demonstrate that our proposed method is able to assign accurate pseudo-labels, and in turn achieves superior classification performance when compared with other existing methods.

## 1 Introduction

Multi-label learning (MLL) is an important learning problem with a wide range of applications (Elisseeff & Weston, 2001; Boutell et al., 2004; Zhang & Zhou, 2006). While traditional setting focuses on the scenario where the label classes are fixed before learning, many real-world applications face different situations. One scenario that is common in many applications is the growing number of classes (Zhu et al., 2018), where the growth splits high-level concepts to finer-grained ones (Deng et al., 2014). For example, the set of classes might start from high-level concepts such as {Animal, . . ., Food }, and then grow to include finer-grained concepts like {Cat, . . ., Dog, . . ., Apple, . . ., Banana}. Typical applications may have collected sufficient number of labeled data for learning the high-level concepts in a fully supervised manner, but it can be challenging for the applications to efficiently adapt the classifier from the high-level (coarse-grained) concepts to the finer-grained ones. Conquering the challenge calls for two components: one is a strategic algorithm to actively collect a few fine-grained and informative labels, and the other is an effective learning model to exploit the fine-grained labels that have been partially collected.

This work focuses on the design of the second component—learning an accurate fine-grained classifier with only limited supervision. In particular, we assume that the model receives a data set that contains all the coarse-grained labels and a few fine-grained ones, as shown in Figure 1a. Then, the problem of constructing a predictive fine-grained model with the presented data set falls under the big umbrella of weakly supervised learning. Specifically, when we focus on leveraging the coarse-grained labels to build a fine-grained classifier, the problem resembles learning with *inexact*

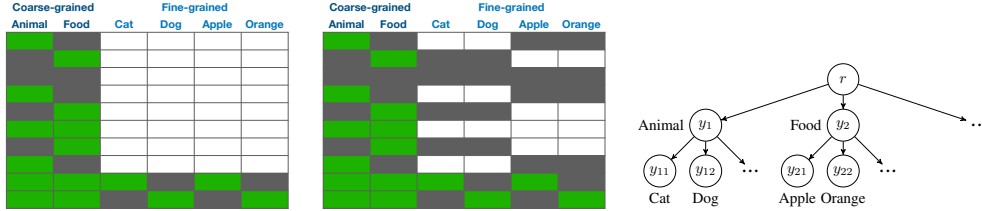

(a) original annotation received (b) deduced from label hierarchy

Figure 1: Each row denotes the label vector of an example, where color green, gray and white correspond to relevant, irrelevant and unknown labels.

Figure 2: Label Hierarchy From Coarse-to-Fine.

*supervision* considered by Zhou (2018), where the coarse-grained labels are not in the exact form for the desired output and could only provide weak information about the target fine-grained labels. On the other hand, if we focus on using the fine-grained part of the labels to train the classifier, the problem can be viewed as a multi-label variant of learning with *incomplete supervision* as some instances receive their exact fine-grained ground-truth labels whereas some do not have labels at all (Zhou, 2018). While both the aforementioned problems have attracted much research attention, the combination of them (inexact and incomplete supervision) which our problem of interest can be cast as, has not yet been carefully investigated to the best of our knowledge.

**Organization** In this work, we start from a formal definition of our problem of interest. We then demonstrate a simple way to reduce the original problem into a special form of negative-unlabeled learning problem (Sakai et al., 2017) leveraging the label hierarchy. To tackle the reduced problem, we begin with a discussion on the caveats carried by some possible existing approaches, and propose a new model that undertakes the challenges posed by inexact and incomplete supervision through a novel learning to learn method which jointly exploits the hierarchical relationship between the coarse- and fine-grained labels, as well as the benefits of all available data in hand. The key idea within our model is to take into account all available information to learn the labeling assignments for the unlabeled entries, called pseudo-labels, and use them to guide the decent direction of the parameter updates on the underlying classifier. Finally, we experimentally demonstrate that the proposed method not only assigns accurate pseudo-labels to the unknown entries but also enjoys significantly better performance than other methods for learning fine-grained classifiers under the limited supervision setting.

## 2 PRELIMINARIES

### 2.1 PROBLEM SETUP

Formally, we denote an instance by a feature vector $\mathbf{x} \in \mathbb{R}^d$, and its relevant labels by a bit vector $\mathbf{y} \in \{-1, 1\}^K$ to indicate whether the labels in a pre-defined set $\mathcal{Y} = \{y_1, ..., y_K\}$ are relevant, i.e., $\mathbf{y}[k] = 1$ if and only if $y_k$ is relevant. In this work, rather than assuming that the set $\mathcal{Y}$ is fixed, we consider the problem of splitting the original high-level concepts into finer-grained ones, refining the label set of interest from $\mathcal{Y}_\mathrm{c} = \{y_1, ..., y_C\}$ into $\mathcal{Y}_\mathrm{f} = \{y_{11}, y_{12}, ..., y_{C1}, y_{C2}, ...\}$ as shown in Figure 2. Let $\mathbf{y}^\mathrm{c}$ and $\mathbf{y}^\mathrm{f}$ be the corresponding label vectors for $\mathcal{Y}_\mathrm{c}$ and $\mathcal{Y}_\mathrm{f}$ respectively. Assume that we receive a data set $\mathcal{D}_\mathrm{tr} = \{(\mathbf{x}_n, \mathbf{y}_n^\mathrm{c})\}_{n=1}^N$ consisting of $N$ examples that are annotated only with the high-level (coarse-grained) labels, and an additional small set $\mathcal{D}_\mathrm{tr}' = \{(\mathbf{x}_m', \mathbf{y}_m'^\mathrm{f})\}_{m=1}^M$ of $M$ examples with their fine-grained labels annotated, our goal is to leverage these examples to learn an accurate *fine-grained* classifier $\Phi(\theta, \mathbf{x}) : \mathbb{R}^d \rightarrow \{-1, 1\}^K$ where $\theta$ is the model parameter and $K$ is the total number of fine-grained classes.

### 2.2 POSSIBLE EXISTING SOLUTIONS

**Fully-Supervised Learning** A straightforward way to learn a fine-grained classifier is to utilize only the fully-annotated training examples in $\mathcal{D}_\mathrm{tr}'$ through standard supervised approaches. Nevertheless, the small number of examples in this set might be unable to train a strong classifier. Moreover, it completely ignores the (weak) supervision provided by the abundant coarse labels.

**Multi-Label Learning with Missing Labels** One way to make use of the higher-level supervision on learning the fine-grained concepts is to leverage the hierarchical relationship where the

Table 1: Precision@K for different methods at different ratios of M/(M+N).

| Ratio (log2) | -10 | | | -9 | | | -8 | | | -7 | | | -6 | | |
|---|---|---|---|---|---|---|---|---|---|---|---|---|---|---|---|
| Methods | P@1 | P@3 | P@5 | P@1 | P@3 | P@5 | P@1 | P@3 | P@5 | P@1 | P@3 | P@5 | P@1 | P@3 | P@5 |
| Standard fully-supervised | 0.5959 | 0.3406 | 0.2487 | 0.6527 | 0.3822 | 0.2802 | 0.6797 | 0.4223 | 0.3127 | 0.7088 | 0.4369 | 0.3220 | 0.7631 | 0.4653 | 0.3387 |
| LEML | 0.4689 | 0.2499 | 0.1785 | 0.6566 | 0.3890 | 0.2883 | 0.6128 | 0.3214 | 0.2252 | 0.6654 | 0.3858 | 0.2810 | 0.7543 | 0.4448 | 0.3211 |
| One-class classification | 0.5515 | 0.2593 | 0.2203 | 0.6105 | 0.3477 | 0.2615 | 0.7051 | 0.4021 | 0.2867 | 0.7408 | 0.4379 | 0.3106 | 0.7564 | 0.4165 | 0.2946 |
| Our method | **0.6633** | **0.4024** | **0.3083** | **0.7217** | **0.4391** | **0.3227** | **0.7410** | **0.4612** | **0.3482** | **0.7705** | **0.4872** | **0.3607** | **0.7792** | **0.4922** | **0.3637** |

irrelevance of a parent (coarse) concept implies the irrelevance of all of its children (fine) concepts. Leveraging the relationship, we are able to infer the corresponding entries in the fine-grained label vectors of the examples in $\mathcal{D}_{\mathrm{tr}}$, making $\mathbf{Y}^{\mathrm{f}} = [\mathbf{y}_1^{\mathrm{f}}, ..., \mathbf{y}_N^{\mathrm{f}}]^\top$ partially observable, and reduce the original problem into a multi-label version of the negative-unlabeled learning problem (Sakai et al., 2017) with very few positive examples, as shown in Figure 1b.

To tackle the reduced problem, one way is to treat the unknown fine-grained labels as missing entries, and apply MLL algorithms that can learn with the presence of missing labels (Goldberg et al., 2010; Xu et al., 2013; Yu et al., 2014). Yu et al. (2014) proposed a classic empirical risk minimization styled method LEML, attempting to solve the optimization problem that arrives at the model parameters which can most accurately recover the observed training labels. Roughly, their objective is formulated as:

$$\theta^* = \arg\min_{\theta} \sum_{(i,j) \in \Omega} L(\Phi(\theta, \mathbf{x}_i)[j], [\mathbf{Y}^{\mathrm{f}}]_{i,j}) + \sum_{m=1}^{M} \sum_{k=1}^{K} L(\Phi(\theta, \mathbf{x}'_m)[k], \mathbf{y}'^{\mathrm{f}}_m)[k]), \tag{1}$$

where $\Omega$ is the set of indices of the observed entries in $\mathbf{Y}^{\mathrm{f}}$ and $L$ is a loss function that measures the discrepancy between the predicted and ground-truth labels. From the objective, however, we note that only the observed entries contribute to the learning of model parameters, and the unobserved ones are basically ignored in the model training process. Unfortunately, in our setting, as the observed fine-grained labels are mostly deduced from the irrelevance of their parent labels, LEML is thus unable to exploit the weak supervision provided by the relevant coarse labels.

**One-Class Multi-Label Learning** Another plausible direction to approach the reduced problem is through one-class multi-label learning methods (Yu et al., 2017). A common approach took in these methods is to assume the values of the unobserved entries to be the opposite class of the observed ones, and train a cost-sensitive classifier with different weights given to the observed and unobserved entries. Nonetheless, as the underlying ground truths for the missing entries are not necessarily the presumed class, without careful algorithm redesign or label distribution estimation, these methods may suffer from the introduced label bias that results in suboptimal performances.

## 3 THE PROPOSED METHOD

While existing solutions have developed different ways of treating the unknown entries during the learning process, they somehow do not delicately exploit the benefits of the unlabeled entries as mentioned in the previous section. In light of this, we seek for a method that could more properly leverage the missing entries with a key assumption that: *When the missing entries are all correctly recovered and used in the training process, the classifier learned could achieve the best performance.* Based on the assumption, we attempt to find the best labeling assignment to the unknown entries, called *pseudo-labels*, which when the model is trained accordingly, can lead to best classification performance on the fine-grained concepts. Towards this goal, we propose to use the few examples that receive their fully-annotated fine-grained labels in $\mathcal{D}'_{\mathrm{tr}}$ as a validation set to evaluate the classifier's performance on the fine-grained concepts. Formally, we aim to train our fine-grained classifier using the examples in $\mathcal{D}_{\mathrm{tr}}$ with a pseudo fine-grained label matrix $\mathbf{Y}^{\mathrm{pseudo}}$ where:

$$[\mathbf{Y}^{\mathrm{pseudo}}]_{i,j} = \begin{cases} [\mathbf{Y}^{\mathrm{f}}]_{i,j} & (i,j) \in \Omega \\ p_{ij} \in [-1,1] & (i,j) \notin \Omega \end{cases}, \tag{2}$$

where every $p_{ij}$ is a pseudo label to be determined and the objective is:

$$\theta^*(\mathbf{Y}^{\mathrm{pseudo}}) = \arg\min_{\theta} \sum_{n=1}^{N} \sum_{k=1}^{K} L(\Phi(\theta, \mathbf{x}_n)[k], [\mathbf{Y}^{\mathrm{pseudo}}]_{n,k}). \tag{3}$$

Note that with different pseudo-labels assigned to the missing entries, we arrive at different optimal model parameter $\theta^*(\mathbf{Y}^{\mathrm{pseudo}})$. And the optimal assignment of the pseudo-labels should be based on the validation performance of the resulting classifier:

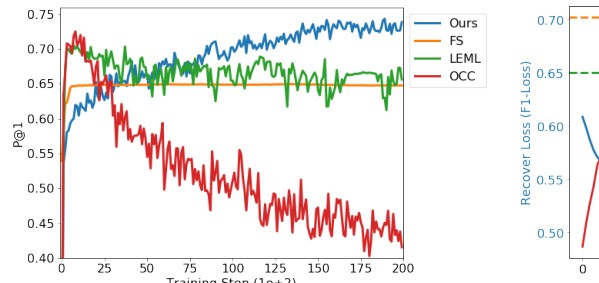

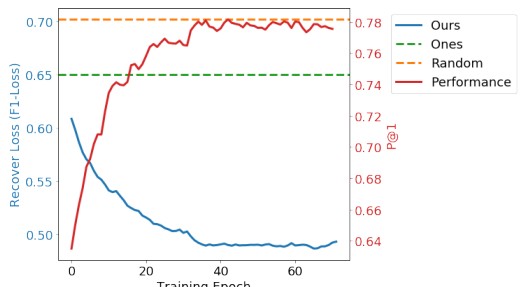

Figure 3: Learning curves of different methods.    Figure 4: Recover rate of the unknown entries.

$$(\mathbf{Y}^{\text{pseudo}})^* = \arg\min_{\mathbf{Y}^{\text{pseudo}}} \sum_{m=1}^{M} \sum_{k=1}^{K} L(\Phi(\theta^*(\mathbf{Y}^{\text{pseudo}}), \mathbf{x}'_m)[k], \mathbf{y}'^{\text{f}}_m[k]). \tag{4}$$

However, solving Eq. 4 to find the optimal pseudo-label assignment requires a computationally prohibiting two-loop optimization procedure. To conquer the optimization challenge, inspired by recent works in meta-learning literature (Finn et al., 2017; Ren et al., 2018), we attempt to tackle the problem with an iterative approach which dynamically find the best pseudo-label assignments locally at each optimization step. Specifically, consider a typical gradient descent update step:

$$\theta^{t+1} = \theta^t - \alpha \nabla \sum_{n=1}^{N} \sum_{k=1}^{K} L(\Phi(\theta, \mathbf{x}_n)[k], [\mathbf{Y}^{\text{pseudo}}]_{n,k}) \Bigg|_{\theta=\theta^t}, \tag{5}$$

where $\alpha$ is the step size and $t$ is the current timestep. Then, at each iteration $t$, we aim to learn the pseudo-label assignment which leads to the model parameters that minimize the validation loss after a single update step:

$$(\mathbf{Y}^{\text{pseudo}})^*_t = \arg\min_{\mathbf{Y}^{\text{pseudo}}} \sum_{m=1}^{M} \sum_{k=1}^{K} L(\Phi(\theta_{t+1}, \mathbf{x}'_m)[k], \mathbf{y}'^{\text{f}}_m[k]). \tag{6}$$

Solving Eq. 6 at each timestep $t$ could, nevertheless, still be very expensive. As a result, we propose a simple approximation of $(\mathbf{Y}^{\text{pseudo}})^*_t$ by looking at the gradient direction (sign) of the validation loss wrt. the pseudo-labels. Particularly, we assign pseudo-labels at timestep $t$ by:

$$[\mathbf{Y}^{\text{pseudo}}_t]_{i,j} = -\text{sign}\left(\frac{\partial}{\partial[\mathbf{Y}^{\text{pseudo}}]_{i,j}} \sum_{m=1}^{M} \sum_{k=1}^{K} L(\Phi(\theta_{t+1}, \mathbf{x}'_m)[k], \mathbf{y}^{\text{f}}_m[k]))\right), \forall(i,j) \notin \Omega. \tag{7}$$

## 4    EXPERIMENTS AND DISCUSSION

To justify the effectiveness of the proposed method, we test our method on a multi-label image dataset MS COCO (Lin et al., 2014). We compare our method with three baseline methods, namely, (1) a standard fully-supervised (FS) learning model, (2) LEML, a classic approach in handling typical missing-label setup (Yu et al., 2014), and (3) a representative method that tackles the problem of one-class classification (OCC) (Yu et al., 2017). We deploy a fully-connected neural network as our underlying base model.

**Comparison with baseline methods**   In Table 1, we show the results of different methods with varying size of $\mathcal{D}'_{tr}$. It can be seen that our method consistently achieves the best performances across different settings. It is worthwhile to note that although the standard fully-supervised approach does not leverage the partially labeled examples in $\mathcal{D}_{tr}$, it surprisingly outperforms the other two baseline methods in many cases. To investigate the reasons for so, we plot the learning curves of different methods in Figure 3. From the figure, we see that although LEML and OCC achieve comparable, or even better, performances than the fully-supervised approach at the very beginning of the learning process, the two approaches then quickly suffers from overfitting that results in the performance drop. For LEML, we conjecture that the performance degrading comes from the overwhelming number of negative entries dominating the learning dynamic. And arguably, the severe overfitting of OCC results from the over-simple assumption on the missing entries which brings label bias into the learning objective.

**Recover rate of our method**   To understand the benefits of the pseudo-labels learned in our approach, we show how our method is capable of correctly recovering the missing entries, as well as the correlation between the recover rate and model performance. In Figure 4, we plot the recover performance of the learned pseudo-labels measured by F1-loss ($1 -$ F1-score), and the horizontal bars are the corresponding F1-loss by simply treating all missing entries as ones and assigning them random labels. We can see from the figure that the pseudo-labels learned from our method could much more correctly recover the missing entries than the two naive baselines. In addition, there is a strong correlation between the recover rate and model classification performance. With more accurate assignment of pseudo-labels on the unknown entries, the trained model is able to achieve stronger classification performance.

**Conclusion**   We design a tailored method through a meta-learning strategy, which learns to accurately assign pseudo-labels to the unknown entries of a special weakly supervised MLL problem. Experimental results show that our proposed method not only assigns accurate pseudo-labels, but also enable the underlying classifier learned to perform better than other possible existing solutions.

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
