# OpenReview forum: "A Pseudo-Label Method for Coarse-to-Fine Multi-Label Learning with Limited Supervision"
_ICLR.cc/2019/Workshop/LLD — LLD 2019_

### Official Review · AnonReviewer2 · 2019-04-04
**Review: A Pseudo-Label Method for Coarse-to-Fine Multi-Label Learning with Limited Supervision**

**Rating:** 2
**Confidence:** 2

**Review:**

The authors propose and implement a new meta-learning approach for multi-label classification where the labels are structured in a two-level hierarchy (i.e. if an example has a label y, it has the label y* where y* is any ancestor of y in the hierarchy).

The text in Figure 2 is very difficult to see since the font size is so small.

I found the use of "relevance" describe a label of y=1 to be a little confusing. Why not use "membership", e.g. y=1 if the given instance belongs to the particular class?

The font in Table 1 is also a little small.

In equation 2, if (i,j) \notin \Omega, then what does [Y^psuedo]_i,j = {0, 1} mean? [Y^psuedo]_i,j isn't set-valued, is it?

In equation 7, the outermost parenthesis should be made larger to more appropriately fit its content. In addition, the sign function typically returns a value in {-1, 0, 1}, but your labels are in {0, 1}.

How did you obtain the feature vector from the examples in MS COCO? What is the classifier you used?

Figure 4 is a bit difficult to read due to the dual y-axes. Perhaps splitting it into two figures would make it easier to understand?

Some grammatical errors detract from the reader's ability to quickly understand the content. For example, there are a number of nouns with missing determiners ("the", "a", etc). Some verbs do not agree with the grammatical number of the subject noun. A few minor spelling errors: "decent" vs "descent", "performances" vs "performance", "recover loss" vs "recovery loss".

---

### Official Review · AnonReviewer1 · 2019-04-08
**This paper investigated an interesting setting, where the coarse labels are abundant and fine-grained labels are scarce.**

**Rating:** 4
**Confidence:** 1

**Review:**


The paper proposed to minimize the discrepancy between inferred fine-grained labels and given coarse labels, by assigning each training example a fine-grained pseudo label, which agrees with the direction of gradient descent. The experiment shows improvement over existing methods.

However, the paper didn't cover the details of the model, which makes their claim less convincing. E.g. does it use hierarchical classification or totally ignore the hierarchical structure in prediction? Is the loss of true labels and pseudo labels weighted?

Also, I think the method of assigning pseudo labels is not quite stable. How about picking the most confident examples in each round, just like self training? It's interesting to see comparison of these methods.

Anyway, as the method and setting are quite novel, I'd recommend acceptance of this paper.

---

### Decision · Program_Chairs · 2019-04-16
**Acceptance Decision**

**Decision:**

Accept

**Comment:**

The paper has some relatively minor issue but an overall interesting concept